# Exploring the Influences of Stream Network Structure and Connectivity on Water Environment Risk in China

Menghan Chen [1,2], Zhicheng Xu [1,3], Lei Cheng [1,2,*], Qinyao Hou [1,2], Pan Liu [1,2] and Shujing Qin [1,2]

1. State Key Laboratory of Water Resources and Hydropower Engineering Science, Wuhan University, Wuhan 430072, China
2. Hubei Provincial Collaborative Innovation Center for Water Resources Security, Wuhan 430072, China
3. Changjiang River Scientific Research Institute, Wuhan 430010, China
* Correspondence: lei.cheng@whu.edu.cn

**Abstract:** Stream networks are the transportation channels of pollutants that can significantly influence water environment risk (WER). However, the influences of stream network structure and connectivity (SC) on WER at the national scale and its regional variability have been rarely investigated in China. In this study, the WER was assessed from the grey water footprint of nitrogen and phosphorus in 214 catchments in China. The relationship between WER and SC and its regional variability were analyzed using correlation and grey relational analysis. Results showed that the water environment risk index ($RI$) in some catchments located in the Hai and Liao River Basins was the highest in China ($RI > 0.8$). On national scale, longitudinal connectivity ($C_l$) and cyclical connectivity ($C_c$) had the strongest influences on WER with grey relational degree index ($GRAI$) of 0.68 and 0.67, respectively. The average slope ($S_r$) was the most important in humid zones, whereas $C_l$ and water surface ratio ($R_w$) had a stronger influence in arid zones. In zones with intensive human activities, $C_c$, river density ($R_d$), and the node connection ratio ($R_{nc}$) mostly affected WER. The main influenced factors varied significantly among nationwide and different zones, which indicated that climate and human activities played an important role in the spatial variation of the relationship between WER and SC. This study highlights the important role of SC on WER and that the relationship between WER and SC varies with climate and human activities.

**Keywords:** water environment risk; stream network structure and connectivity; regional variability; climate; human activity intensity; grey relational analysis

## 1. Introduction

With the acceleration of urbanization and rapid industrial and agricultural development, a large number of pollutants are discharged into river channels and water bodies [1–4], which increases the water environment risk (WER). Similar to most developing countries, water pollution (e.g., surface water eutrophication and water quality decline) is one of the most critical environmental problems in China with rapid urbanization and economic development [5,6]. Over the past few decades, sewage discharge in China has shown an increasing trend [7], and water quality in 20.6% of surface water falls into class V or worse according to statistics [8]. Rivers, lakes, and other water bodies form a complex and interconnected stream network in China, which is related to the WER due to the water cycle, pollutant transport, and biogeochemical processes [9,10]. Generally, point source pollution directly discharged into rivers and non-point source pollution driven by rainfall-runoff and discharged into stream networks are mainly collected and transported by rivers, and the transfer and dilution of pollutants are affected by the stream network structure and connectivity (SC) [11,12]. Several studies confirmed the impact of SC on WER in some catchments [10,11,13]. However, the spatial variations in WER and its response to SC at the national scale in China have rarely been investigated.

Numerous studies showed that the number, morphology, and connectivity characteristics of stream networks and topography significantly impact the catchment water quality [10,11,14]. For example, Yu et al., revealed that the water surface ratio played a key role in affecting water quality in the Yangtze River Delta plain [10]. Zhao et al., indicated that river density had a positive correlation with water quality in Shanghai [15]. Helton et al. found that the shape of the stream network could affect the removal of $NO_3^-$ [16]. Lyu et al., stated that the concentration and distribution of phosphorus were affected by river connectivity in Jiangsu Province [9], and a case in America reported that water quality parameters of most dry seasons were related to topographic features such as slope [17]. Janardan pointed out the influences of mean slope and water surface ratio on total phosphorus at different scales in the Han River basin, South Korea [18], Carlson Mazur stated the impact of hydrologic connectivity on water quality in the Wabash-White watershed [19]. However, in earlier studies, the main driving stream network structure and connectivity characteristics (SCCs) that affected water quality were unclear and dissimilar in different catchments. For example, river length was the key factor affecting total organic carbon (TOC) in the Nakdong River basin [20]; Knoll et al., found that elevated trophic status in Ohio lakes in the US were associated with shallow lake depth and high catchment-to-water surface area ratio [21]. In China, Dou et al., found that the node connection rate and river density had the greatest impact on the water quality of Zhengzhou City [13]; the water surface ratio was the main factor affecting water quality in the Southern Jiangsu Plain and Yangtze River delta plain while the influence of the node connection ratio in these two areas was relatively weak [10,11,14]. Thus, identifying the main driving SCCs at the national scale while elucidating regional differences in the main driving SCCs can provide a sound understanding of the relationship between WER and SC. However, such investigations have rarely been conducted at the national scale.

In addition to SC, climate, human activities, land-use types, and topography are also important driving factors affecting water quality [22–26]. Jiang et al., investigated the relationship between climate and river water quality on a global scale, using the elasticity approach [27]. Kaushal et al., summarized the origins, evolution, and resilience of diverse water quality responses to extreme climate events [28]. Liu et al., reported that human activities significantly affected the water quality of the Hongze Lake [23]. Nguyen stated the pollution source of human activities (industrial, agricultural, and residential sources) directly affected surface water quality in southwestern Vietnam [29]. Yu et al., and You et al., found that land-use type and topography also strongly influenced water quality [26,30]. Meanwhile, SCCs are also affected by climate and human activity [31–34]. Kreiling found that both water body area and number were affected by climate change and anthropogenic water exploitation in the Fox River watershed [35]. Ranjbar et al., revealed the systematic impacts of climate forcing on the stream network topology and geometry in basins with drainage density in America [36]. Luo et al., quantified the relationship between the human activity intensity and evolution of the SC in the Shaying River basin [31]. Therefore, climate and human activities may also have confounding effects on the correlation between WER and SC. These confounding influences may not be important at small spatial scales or for specific cases but should be considered at large spatial scales with significant spatial differences in climate and human activities.

Therefore, the overall objective of this study is to explore the spatial variability of the relationship between WER and SC under the impact of climate and human activities. This study analyzed the relationship between WER and SC at a large spatial scale and its spatial difference was revealed by grouping level-III catchments according to climate and human activity intensity. In this study, the WER was estimated using the water pollution levels of nitrogen and phosphorus at the catchment scale, and the relationship between WER and SC was investigated using the correlation analysis method and grey relational analysis at the national and regional scales. The specific objectives of this study were to: (1) estimate the WER and calculate SCCs in China at the level-III catchment scale; (2) explore the relationship between WER and SC at a national scale; and (3) explore the

regional variabilities in the relationship between WER and SC by further considering the impact of climate and human activities.

## 2. Materials and Methods

### 2.1. Study Area

China is a vast country with wide latitude and different mountain orientations, and has a varied topography including mountainous regions, plateaus, basins, plains, and hills. China has a complex climate with monsoon, temperate continental, and alpine climates in the east, northwest, and on the Qinghai-Tibet Plateau, respectively. The total volume of water resources, including many rivers and lakes, is abundant in China, which include more than 1500 rivers with a drainage area above 1000 km$^2$, more than 2800 natural lakes with an area above 1 km$^2$, and a large number of dams and hydropower stations, forming a complex stream network. China is divided and sub-divided into successively smaller water resource regions which are classified into three levels: level-I (Figure 1b), level-II, and level-III (Figure 1a) catchments. The 10 level-I catchments are divided according to the integrity of major river basins in China, and the 80 level-II and 214 level-III catchments are divided according to the distribution of tributary river systems and the opinions of local management departments [37]. The area of the level-III catchments was ranging from $2.98 \times 10^3$ to $7.03 \times 10^5$ km$^2$, and mainly concentrated at $2.00 \times 10^4 \sim 5.00 \times 10^4$ km$^2$. The values of WER and SCCs were calculated at the level-III catchment scale in this study.

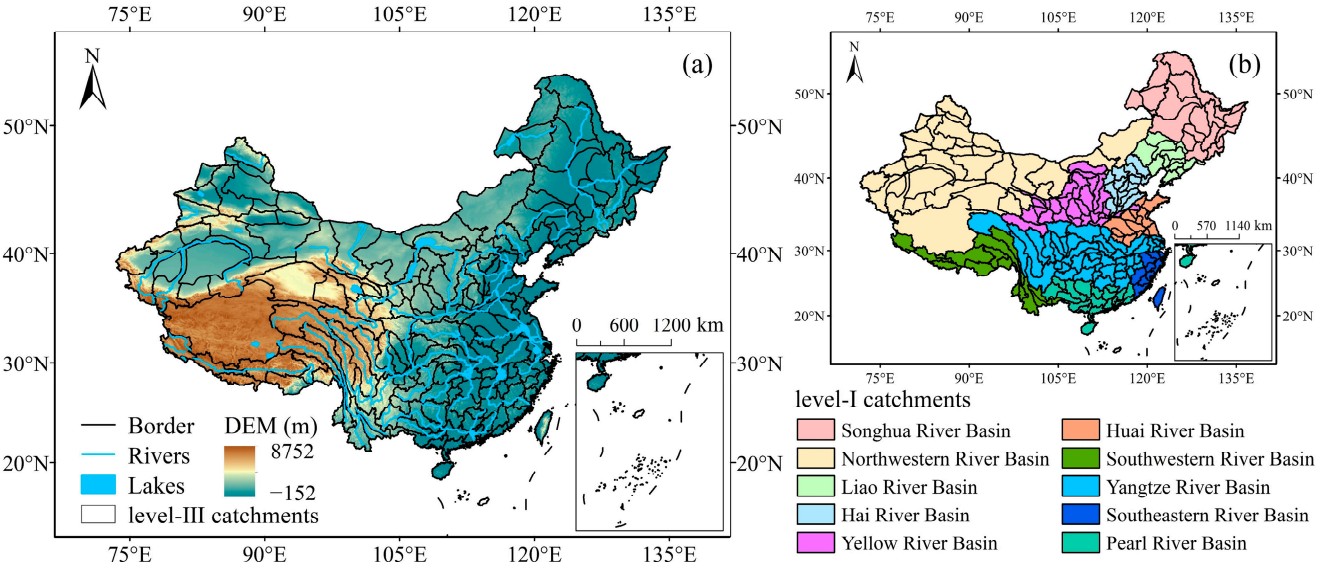

**Figure 1.** Map of the study area. The subplot (**a**) shows the main rivers, lakes, level-III catchments and the digital elevation model (DEM), and (**b**) shows the 10 level-I catchments in China.

Precipitation and human activity intensity have strong spatial variability in China (Figure 2). The mean annual precipitation of each catchment was calculated by the area-weighted average method using the China Meteorological Forcing Dataset (CMFD) with a spatial resolution of 0.1° × 0.1° [38]. The mean annual precipitation in level-III catchments ranged from 75 to 3340 mm and decreased from the southeast to the northwest. The human activity intensity index data were comprehensively calculated from several aspects of human influence, including population density, land transformation, accessibility, and electrical power infrastructure, at a spatial resolution of 1 km × 1 km. The human activity intensity index data were jointly produced by the Center for International Earth Science Information Network (CIESIN) at Columbia University and the Wildlife Conservation Society [39], and was area-weighted and averaged to the level-III catchment scale. It ranged from 2.8 to 50.6. Human activities in North China were the most intense, while the Qinghai-Tibet Plateau and Northwest China were less affected by human activities.

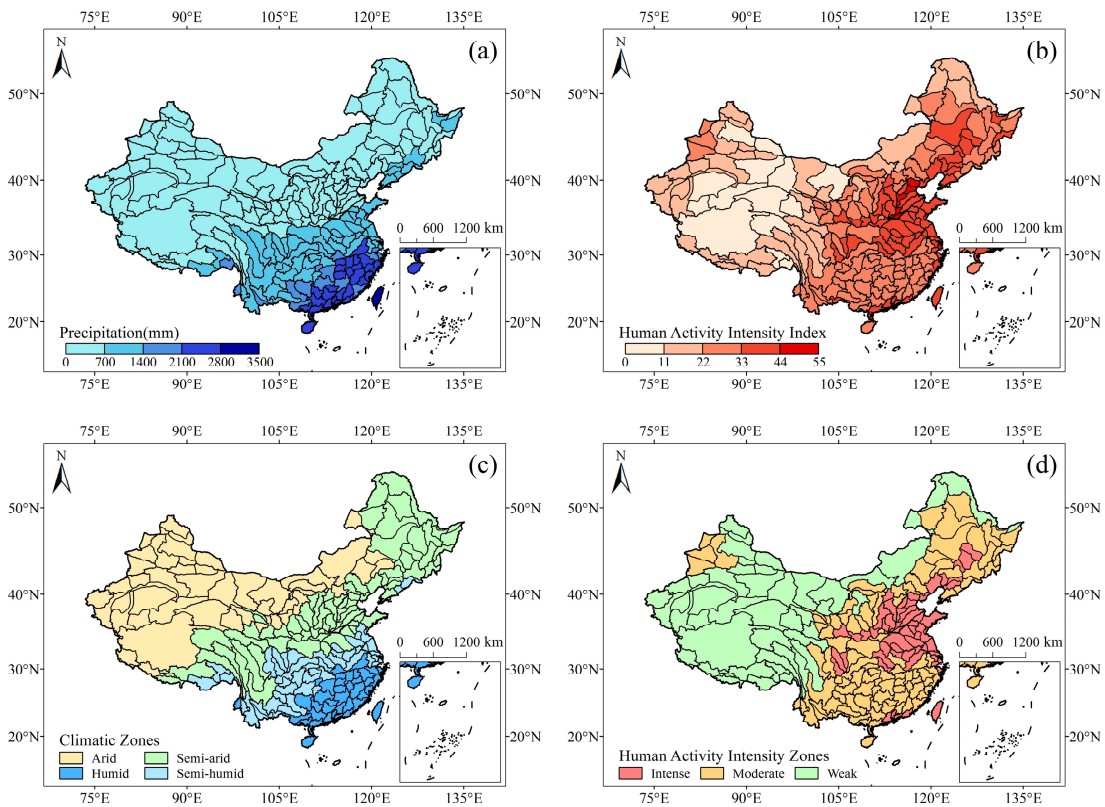

**Figure 2.** Maps depicting the spatial variations of (**a**) mean annual precipitation, (**b**) human activity intensity, and their clustered zones determined by the K-means clustering algorithm in (**c**) and (**d**), respectively.

### 2.2. Proposed Water Environment Risk Index (RI)

In this study, WER was assessed based on the water pollution level (WPL). The WPL is the ratio of grey water footprint (GWF) of pollutants to local river runoff and was widely used to assess regional water pollution status [40–42]. GWF was developed to evaluate WPL and was applied in many fields [43], such as agricultural, industrial, domestic, and GWF flows between regions [44]. WPL of nitrogen ($N_{wpl}$) and phosphorus ($P_{wpl}$) were selected to assess catchment water pollution degree (*WP*), as nitrogen and phosphorus are the two main pollutants affecting water quality [45–47] and high-resolution pollutant records of nitrogen and phosphorus are available nationwide. The *WP* was reflected by the maximum values of $N_{wpl}$ and $P_{wpl}$ and was calculated as follows:

$$WP = \max(N_{wpl}, P_{wpl}) \tag{1}$$

$$N_{wpl} = \frac{N_{gwf}}{\overline{R}} \tag{2}$$

$$P_{wpl} = \frac{P_{gwf}}{\overline{R}} \tag{3}$$

where $N_{gwf}$ and $P_{gwf}$ are the catchment grey water footprints of nitrogen and phosphorus with units of mm/yr, respectively, and $\overline{R}$ is the annual mean runoff with units of mm/yr. $N_{gwf}$ and $P_{gwf}$ were obtained from Mekonnen and Hoekstra [41] and Mekonnen and Hoekstra [42], respectively, which were estimated at a spatial resolution of 0.083° × 0.083° for the period 2002–2010 using multi-source data (e.g., crop distributions from Chad et al. [48], and the rate of mineral fertilizer applied to major crops from IFA, et al.). $\overline{R}$ is the gridded runoff at a spatial resolution of 0.5° × 0.5° for the period 2002–2010, and an optimal weighting approach was applied to merge runoff estimated from eight hydrological models constrained

by observational streamflow records [49]. Datasets with different spatial resolutions were rasterized to $0.5° \times 0.5°$ resolution using ArcGIS 10.7 software to calculate the grid *WP*, and then the area-weighted average method was used to calculate the catchment scale *WP*.

The extreme value standardization method was used to scale *WP* to 0–1.0. The standardized *WP* was used as the proposed *RI*, and was calculated as follows:

$$R\,I = \frac{WP - WP_{min,95}}{WP_{max,95} - WP_{min,95}} \tag{4}$$

where $WP_{min,95}$ and $WP_{max,95}$ are the minimum and maximum 95% fractals of all catchment *WP*, respectively. Equation (4) considers the possible influence of outliers caused by the data uncertainty. This method can prevent one or a few outliers from having a dramatic impact on the scaling of all other data [50,51].

### 2.3. Stream Network Structure and Connectivity Characteristics (SCCs)

Based on earlier studies, six SCCs, namely river density ($R_d$), water surface ratio ($R_w$), average slope ($S_r$), longitudinal connectivity ($C_l$), cyclical connectivity ($C_c$), and node connection ratio ($R_{nc}$), were selected to explore their potential effects on WER. Details of the calculation equations, implications for water quality, and references of these six SCCs are shown in Table 1. $R_d$ and $R_w$ are important characteristics of river magnitude that can reflect the level of pollution transfer and the capacity to carry pollutants [11]. $S_r$ is a topographic feature that has a significant impact on water quality by affecting water movement [25,52]. $C_l$, $C_c$, and $R_{nc}$ are commonly used to describe the connectivity of stream networks, which could affect the flow of water upstream and downstream, flow paths, and the possibility of pollution transfer between each stream node and its surrounding streams [11,13,31,53] and thus affecting the WER. The above six SCCs indexes were calculated at level-III catchment scale using ArcGIS 10.7 software based on publicly available datasets, including the digital elevation model (DEM) at 30-m resolution (http://www.geodata.cn (accessed on 20 August 2020)), a high-resolution (30 m × 30 m) global surface water dataset [54], main rivers in China from the National Earth System Science Data Center of China (http://www.geodata.cn (accessed on 9 September 2020)), and the Global Geo-referenced Database of Dams (GOODD) from the Global Dam Watch (http://globaldamwatch.org/data/ (accessed on 29 August 2021)).

**Table 1.** Stream network structure and connectivity characteristics (SCCs).

| Indexes | Formula | Implication for Water Quality | Reference |
|---|---|---|---|
| River density | $R_d = \frac{L}{A}$ | Level of pollution transfer of the river channels | [11] |
| Water surface ratio | $R_w = \frac{A_w}{A} \times 100\%$ | Capacity of the rivers and lakes to carry pollutants | [11] |
| Average slope | $S_r = \frac{\sum_{i=1}^{Nr} S_i}{N_r}$ | Movement and decomposition of pollutants by affecting water flow | [25] |
| Longitudinal connectivity | $C_l = \frac{N_d}{L_r}$ | Transmission of flow and pollutants upstream and downstream | [55] |
| Cyclical connectivity | $C_c = \frac{N_r - N_p + 1}{2N_p - 5}$ | Optional degree of the moving routes of pollutants | [13] |
| Node connectivity ratio | $R_{nc} = \frac{N_r}{N_p}$ | Possibility of pollution transfer between the two nodes | [11] |

Note: $R_d$ is the river density, km/km$^2$; $L$ is the total length of rivers, km; $A$ is the total area of a catchment, km$^2$; $R_w$ is the water surface ratio; $A_w$ is the total area of the rivers and lakes under the mean water level in a catchment, km$^2$; $S_r$ is the average slope, degree; $S_i$ is the slope of the *i*-th river, degree; $N_r$ is the total number of the rivers; $C_l$ is the longitudinal connectivity, km$^{-1}$; $N_d$ is the number of the dams and reservoirs; $L_r$ is the length of a river, km; $C_c$ is the cyclical connectivity; $N_p$ is the number of nodes; $R_{nc}$ is the node connectivity ratio. The statistics and calculation of all variables were made at the level-III catchments as the basic spatial unit using ArcGIS 10.7 software.

### 2.4. Analysis Methods

Correlation analysis method is widely used to reveal the relationship between influence factors and water quality [26,56]. The Pearson correlation coefficient ($r$) was used to reflect the relationship between WER and SCCs in this study, which was calculated as:

$$r = \frac{\sum_{i=1}^{n}(x_i - \bar{x})(y_i - \bar{y})}{\sqrt{\sum_{i=1}^{n}(x_i - \bar{x})^2 \sum_{i=1}^{n}(y_i - \bar{y})^2}} \tag{5}$$

where $x_i$ is the $i$-th evaluation variable sequence, $y_i$ is the $i$-th corresponding variable sequence, $\bar{x}$ is the mean value of the evaluation variable sequence, $\bar{y}$ is the mean value of the corresponding variable sequence, and $n$ is the total number of variables in each variable sequence.

Grey relational analysis (GRA) can solve the correlation problem of grey systems with incomplete data [11]. According to grey system theory, a grey system refers to a system in which part of the information is known and a part is unknown [57,58]. A system of water environment risk and stream network structure and connectivity characteristics can be regarded as a grey system. The GRA reveals the nonlinear relationship between variables according to the similarity of multivariable geometric proximity, quantitatively estimates the geometric proximity between data sequences, and ranks their relational degrees in descending order [59,60]. This study used GRA to analyze the relationship between WER and SC. The grey relational degree index (*GRAI*) was calculated as follows [59,61]:

(1)  Determine the reference and comparison sequences.

The reference sequence was set as:

$$X_0 = \{x_0(i) \mid i = 1, 2, \ldots, n\} \tag{6}$$

where $x_0(i)$ represents the *RI* of the $i$-th level-III catchment and $n$ represents the number of level-III catchments.

The comparison sequences were set as:

$$X_k'(i) = \{x_k(i) \mid k = 1, 2, \ldots, m; \ i = 1, 2, \ldots, n\} \tag{7}$$

where $x_k(i)$ represents the value of the $k$-th SCCs of the $i$-th level-III catchment and $m$ represents the number of SCCs.

(2)  Normalize the reference and comparison sequences.

Normalization is typically required because the range and units are often different for each sequence. Various methods were proposed for the normalization of GRA, and in this study the sequences were normalized as follows:

$$X_k(i) = \frac{X_k'(i) - \min X_k'(i)}{\max X_k'(i) - \min X_k'(i)} \tag{8}$$

$$X_k(i) = \frac{\max X_k'(i) - X_k'(i)}{\max X_k'(i) - \min X_k'(i)} \tag{9}$$

where $X_k'(i)$ and $X_k(i)$ represent the original and normalized sequences, respectively. According to the positive and negative correlations between the reference and comparison sequences, the reference and positive comparison sequences were normalized using Equation (8), while the negative comparison sequences were normalized using Equation (9).

(3)  Determine the deviation sequences.

$$\Delta_k(i) = |x_0^*(i) - x_k^*(i)| \tag{10}$$

where $\Delta_k$ represents the deviation sequences between the corresponding values in the reference and $k$-th comparison sequences, $x_0^*(i)$ and $x_k^*(i)$ represent the normalized values of the reference and comparison sequences, respectively.

(4)     Determine the grey relational coefficient.

$$\xi_k(i) = \frac{\min\Delta_k(i) + \rho\max\Delta_k(i)}{\Delta_k(i) + \rho\max\Delta_k(i)} \tag{11}$$

where $\rho \in [0, 1]$ is the distinguishing coefficient that differentiates the degree of proximity of the reference and comparison sequences, such that $\xi_k(i) \in [0, 1]$. Generally, $\rho = 0.5$ is widely set based on the minimum information [59].

(5)     Calculate the grey relational degree index.

$$GRAI_k = \frac{1}{n}\sum_{i=1}^{n} \xi_k(i) \tag{12}$$

where $GRAI_k$ represents the $GRAI$ between the reference and $k$-th comparison sequences.

### 2.5. K-Means Clustering Algorithm

The K-means clustering algorithm is a numerical, unsupervised, non-deterministic, and iterative method for dividing a dataset into a certain number of subsets, and was first proposed by MacQueen [62]. It is a partitioning clustering algorithm that classifies given data objects into $k$ different clusters through iterative convergence to a local minimum. This method is simple and fast; therefore, it is widely used in many practical applications [63]. As shown in Figure 2, level-III catchments were grouped into four climatic zones (i.e., humid, semi-humid, semi-arid, and arid zones) and three human activity intensity zones (i.e., intense, moderate, and weak zones) according to the mean annual precipitation [64,65] and human activity intensity index using this method, respectively. The cluster values of the different zones are listed in Table 2.

**Table 2.** Cluster values of precipitation (mm) in different climatic zones and human activity intensity index in different human activity intensity zones.

| Climatic Zones | | Human Activity Intensity Zones | |
|---|---|---|---|
| **Zones** | **Values** | **Zones** | **Values** |
| Humid | 1797 | Intense | 40 |
| Semi-humid | 1158 | Moderate | 29 |
| Semi-arid | 624 | Weak | 13 |
| Arid | 251 | - | - |

## 3. Results

### 3.1. Spatial Distribution of WER and SCCs in China

The spatial distribution of the $RI$ in China is shown in Figure 3. The spatial variability of the WER in China is very obvious. Regions with high $RI$ were mainly distributed in North China, which was notably higher than that in other parts of China. The WER in the Hai and Liao River Basins was the highest, with $RI$ generally reaching more than 0.7, especially in the eastern parts of the Hai River Basin ($RI > 0.9$). Simultaneously, catchments in the Songhua River Basin, northwestern China, southern China, and Qinghai-Tibet Plateau had a low WER ($RI < 0.4$). Furthermore, the two catchments in the Northwestern River Basin had a relatively high WER compared with that of the surrounding areas. Overall, the WER was low, except in North China.

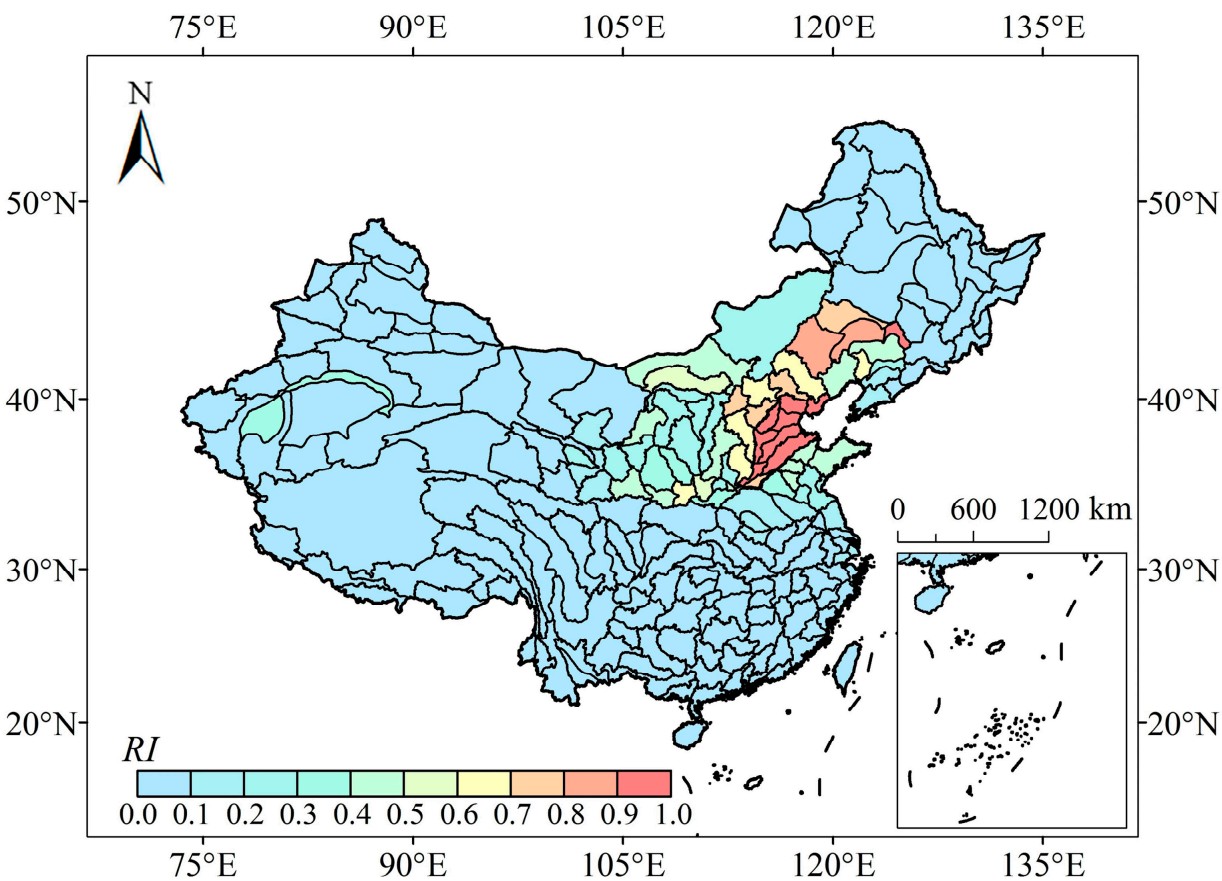

**Figure 3.** Map of the spatial distribution of water environment risk index (*RI*) in China.

　　　Figure 4 shows the spatial distributions of the six SCCs. $R_d$ increased from 0.00–0.05 km/km$^2$ in the northwest inland to above 0.09 km/km$^2$ in Southeast China, among which the $R_d$ in the estuary of the Pearl River and Yangtze River Deltas attained greater than 0.09 km/km$^2$ (Figure 4a). $R_w$ was higher on the east coast, especially in the plain area of the lower reaches of the Yangtze River, and in lake distribution areas such as the Qinghai Lake and the hinterland of the Qinghai-Tibet Plateau (Figure 4b). $R_d$ and $R_w$ have similar spatial distribution characteristics in the southeast China. The values of $S_r$ ranged from 0.0° to 13.2° in China and were the largest at the edge of the Qinghai-Tibet Plateau, while those were smaller in North China, Central China, the Northeast Plain, and the Middle-Lower Yangtze River Plain (Figure 4c). The longitudinal connectivity in the south was significantly weaker than that in the north, especially in the south of the Yangtze River Basin, with a maximum $C_l$ value of 1.76/100 km, which indicated the weakest longitudinal connectivity (Figure 4d). Simultaneously, the $C_l$ in the Yellow River Basin showed a spatial difference. The $C_c$ and $R_{nc}$ showed no significant regional differences in China and were relatively large in the Yellow River Basin, the Middle-Lower Yangtze River, and the estuary of the Yangtze and Pearl Rivers (Figure 4e,f) with the range, mean value, and variance of $C_c$ and $R_{nc}$ being 0.00–0.97, 0.30 and 0.18, 0.00–2.15, 1.11 and 0.47, respectively. Generally, $R_d$, $R_w$, and $C_l$ showed high values in the east and low values in the west, whereas $S_r$ showed a decreasing trend from the southwest to northeast. The spatial distributions of $C_c$ and $R_{nc}$ were relatively uniform in China. Additionally, the values of all six SCCs in some catchments in the western Northwest China were relatively high compared to that of the surrounding regions.

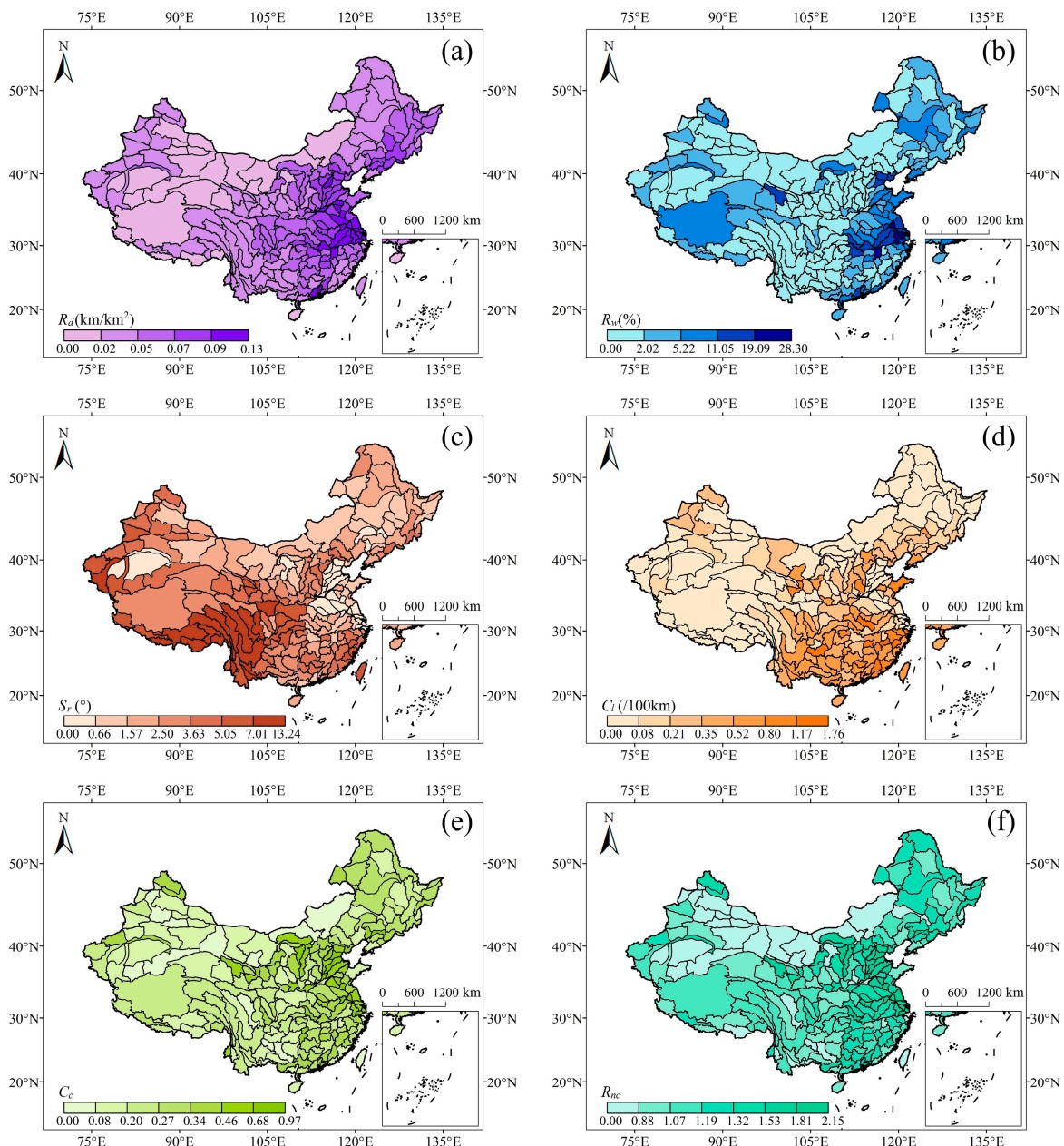

**Figure 4.** Maps of the spatial distributions of (**a**) river density ($R_d$), (**b**) water surface ratio ($R_w$), (**c**) average slope ($S_r$), (**d**) longitudinal connectivity ($C_l$), (**e**) cyclical connectivity ($C_c$) and (**f**) node connection ratio ($R_{nc}$) in China.

### 3.2. Correlation between WER and SC

The relationship between WER and SC was relatively complex, with both positive and negative Pearson correlation coefficients (Figure 5). At the national scale, $S_r$ and $C_l$ were negatively correlated with WER, especially $S_r$, which had the strongest correlation ($r = -0.39$). Conversely, $R_d$, $C_c$, and $R_{nc}$ were positively correlated with WER, with $C_c$ having the strongest correlation ($r = 0.34$). Furthermore, $R_w$ and $C_l$ showed a non-significant correlation with WER, and the correlation between $R_w$ and WER was very weak ($r = 0.02$). Generally, the results of the correlation analysis showed that the six SCCs had a certain impact on WER, but the impact was limited at the national scale.

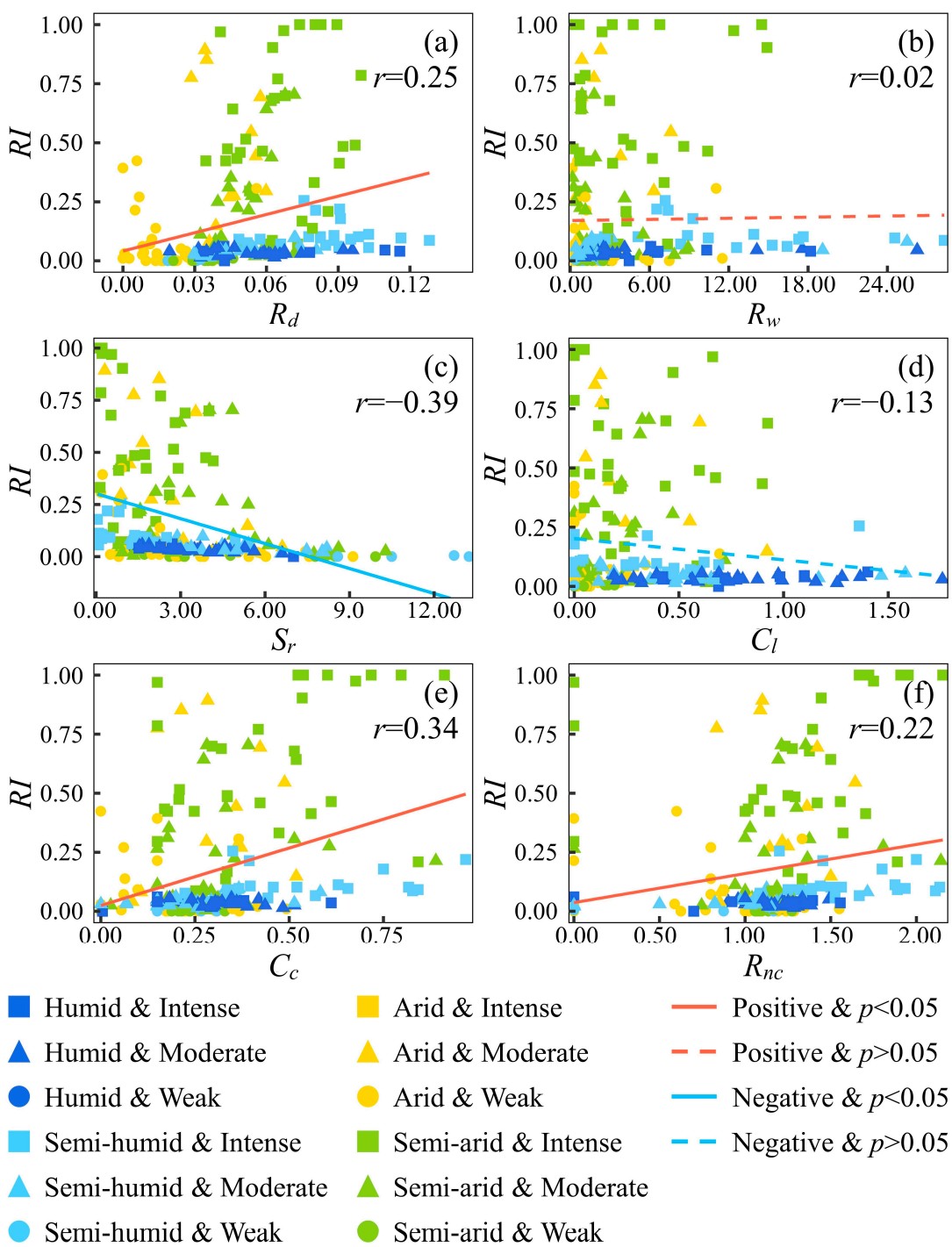

**Figure 5.** Plots illustrating the correlation between WER and SCCs in China. The subplots (**a–f**) showed correlations between $RI$ and $R_d$, $RI$ and $R_w$, $RI$ and $S_r$, $RI$ and $C_l$, $RI$ and $C_c$, $RI$ and $R_{nc}$, respectively.

GRA was applied to determine the quantitative relationship between WER and SCCs and to further reveal the relationship between WER and SC. According to the results of the cluster analysis, the $GRAI$ between WER and SCCs was calculated for different climate and human activity intensity zones (Figure 6), and the average $GRAI$ was calculated for each SCC nationwide (Table 3). Figure 6 and Table 3 show that the $GRAI$ between WER and the six SCCs in each climatic and human activity intensity zone and nationwide was generally greater than the distinguishing coefficient ($\rho = 0.5$), which indicated that there

was a strong correlation between WER and SC. Additionally, at the national scale, the *GRAI* between *RI* and each SCC had small difference (ranging from 0.58 to 0.68) and could be sorted as $C_l > C_c > R_d = R_w > R_{nc} > S_r$ according to the *GRAI*. The $C_l$ and $C_c$ had relatively the strongest correlation with WER compared with other SCCs, and the *GRAI* were 0.68 and 0.67, respectively. $R_d$ and $R_w$ had an equivalent degree of correlation with WER nationwide (*GRAI* = 0.64), while $R_{nc}$ and $S_r$ showed a relatively weak correlation. These results indicate that the WER was evenly related to the SC, and the effect of $C_l$ and $C_c$ on WER was stronger than those of the other SCCs at the national scale.

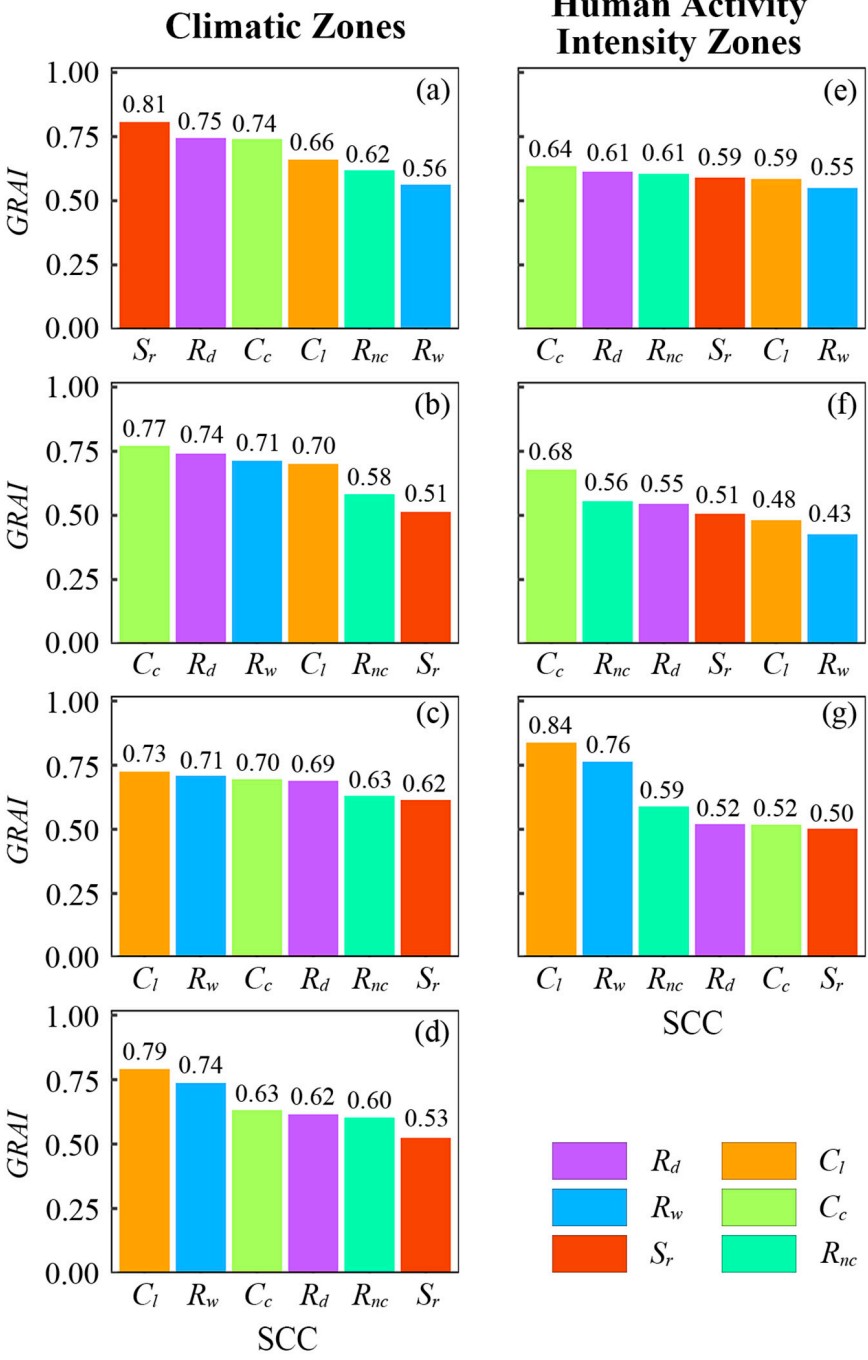

**Figure 6.** Plots illustrating the grey relational degree index (*GRAI*) in different climatic zones (left panel) and human activity intensity zones (right panel). The subplots in left panel (**a**–**d**) showed *GRAI* of six SCCs in humid, semi-humid, semi-arid and arid zones, respectively. The subplots in right panel (**e**–**g**) showed *GRAI* of six SCCs in intense, moderate and weak zones, respectively.

**Table 3.** The *GRAI* between *RI* and each SCC nationwide.

|  | $R_d$ | $R_w$ | $S_r$ | $C_l$ | $C_c$ | $R_{nc}$ |
|---|---|---|---|---|---|---|
| *GRAI* | 0.64 | 0.64 | 0.58 | 0.68 | 0.67 | 0.60 |

### 3.3. Regional Variability of the Relationship between WER and SC

The correlation between WER and SC increased and showed a great difference when climatic conditions and human activity intensity were considered (Tables 4 and 5). For example, the *r* of $S_r$ increased to more than 0.66 in the humid and semi-humid zones. $R_w$ was significantly correlated with WER in different zones, in contrast to the national scale, whereas the correlation of $C_l$ was still non-significant. Simultaneously, $R_d$, $R_w$, and $C_l$ showed opposite correlations in climatic and human activity intensity zones. These results indicate that climate and human activity significantly influenced the relationship between WER and SC.

**Table 4.** Pearson correlation coefficient (*r*) between WER and SCCs in different climatic zones.

| Climatic Zones | *r* | | | | | |
|---|---|---|---|---|---|---|
|  | $R_d$ | $R_w$ | $S_r$ | $C_l$ | $C_c$ | $R_{nc}$ |
| Humid | 0.14 | 0.42 * | −0.74 * | 0.06 | −0.06 | −0.08 |
| Semi-humid | 0.61 * | 0.34 * | −0.66 * | 0.01 | 0.59 * | 0.55 * |
| Semi-arid | 0.54 * | 0.26 * | −0.46 * | 0.16 | 0.53 * | 0.29 * |
| Arid | 0.35 * | 0.03 | −0.42 * | 0.22 | 0.21 | 0.24 |
| Nationwide | 0.25 * | 0.02 | −0.39 * | −0.13 | 0.34 * | 0.22 * |

Note: * indicate $p < 0.05$.

**Table 5.** Pearson correlation coefficient (*r*) between WER and SCCs in different human activity intensity zones.

| Human Activity Intensity Zones | *r* | | | | | |
|---|---|---|---|---|---|---|
|  | $R_d$ | $R_w$ | $S_r$ | $C_l$ | $C_c$ | $R_{nc}$ |
| Intense | −0.15 | −0.31 * | −0.22 | −0.30 * | 0.17 | 0.04 |
| Moderate | −0.04 | −0.10 | −0.27 * | −0.24 * | 0.15 | 0.14 |
| Weak | −0.29 * | 0.03 | −0.45 * | 0.07 | −0.27 | −0.20 |
| Nationwide | 0.25 * | 0.02 | −0.39 * | −0.13 | 0.34 * | 0.22 * |

Note: * indicate $p < 0.05$.

The results of the GRA also confirmed the influence of climate and human activities on the relationship between WER and SC. As shown in Table 6, the WER in different zones differed from the average *GRAI* of the six SCCs. The average grey relational degrees could be sorted as humid > semi-arid > semi-humid > arid in climatic zones; however, the value of *GRAI* showed a small difference. Similarly, the average grey relational degrees could be sorted as weak > intense > moderate in the human activity intensity zones, which indicated that the correlation between WER and SC may first decrease and then increase with an increase in human activity intensity.

**Table 6.** Average *GRAI* of the different climatic and human activity intensity zones.

| | Climatic Zones | | | | Human Activity Intensity Zones | | |
|---|---|---|---|---|---|---|---|
| | Humid | Semi-Humid | Semi-Arid | Arid | Intense | Moderate | Weak |
| *GRAI* | 0.69 | 0.67 | 0.68 | 0.65 | 0.60 | 0.53 | 0.62 |

The main SCCs that affected the WER were different in the different zones (Figure 6). For different climatic zones (Figure 6a–d), $S_r$ had the strongest grey relational degree in

humid zones and the weakest in other climatic zones. $R_d$ and $C_c$ showed a relatively strong correlation in humid and semi-humid zones (*GRAI* > 0.74), while $R_w$ was more important in semi-humid zones. The correlation between $R_{nc}$ and WER was weak in all four climatic zones. The order of importance of different SCCs on WER was the same in semi-arid and arid zones; furthermore, the difference in grey relational degree between each characteristic and the WER of arid zones was more obvious.

Similarly, the main SCCs also differed across human activity intensity zones (Figure 6e–g). $C_c$, $R_d$, and $R_{nc}$ were relatively important in the intense and moderate zones, whereas $Sr$, $C_l$, and $R_w$ had a low grey relational degree, especially $C_l$ and $R_w$ in moderate zones. Simultaneously, the *GRAI* of each SCC exhibited minimal differences in the intense zones (ranging from 0.55 to 0.64). Contrastingly, the *GRAI* of each SCC varied greatly in the weak human activity intensity zone. Compared with the other SCCs, $C_l$ showed the strongest correlation (*GRAI* = 0.84), followed by $R_w$ (*GRAI* = 0.76). Summarily, the relationship between WER and SC showed regional variability under the influence of climate and human activity.

## 4. Discussion

### 4.1. Spatial Variability of WER in China

In this study, the WPL was used to reflect the WER at the catchment scale. The results showed that the WER had an obvious spatial variability in China. This result is reasonable because China also has significant spatial differences in terms of climate, human activities, stream network systems, industrial and agricultural development, etc. Several studies reported spatial differences in water pollution levels and risks. Compared with other parts of China, North China and some parts of Northeast China have a higher water pollution risk and greater water pollution pressure [66–68]. Cai et al., found that provinces located on the northern and central coasts had relatively higher water pollution levels [66]. Ouyang et al. reported a high potential risk of pesticide pollution in Henan, Shandong, Hebei, and Beijing [69]. Wang et al., stated that river pollution was spatially uneven and clustered in China, and the most serious water pollution occurred in the Huai, Hai, Yellow, and Liao River Basins [67].

Although only two common water pollutants (i.e., nitrogen and phosphorus) were used to evaluate WER in this study, the spatial distribution of WER was similarly mapped with those in earlier studies that used the annual provincial data of total wastewater discharge with more than 10 pollutants [66]. Compared with earlier studies, this study carried out a reasonable and reliable nationwide WER with higher resolution, which provided an opportunity to explore the relationship between WER and SCCs.

### 4.2. Complex Relationship between WER and SC

The results of the correlation analysis and GRA suggest that the relationship between WER and SC is relatively close in China. Earlier studies showed that SC had a significant impact on water quality [10,11,19,70]. Generally, pollutants are collected by tributaries, and transported, diluted, and deposited in streams and lakes [71], and such capability is highly related to the structure and connectivity of regional and/or large-scale stream network [11]. On the one hand, water bodies have the capacity of carrying and diluting pollutants, and the concentrations of pollutants can be changed by a series of physical, chemical, and biological processes within water bodies [14]. When the pollutant load exceeds the self-purification capacity of the water body, the water quality deteriorates, resulting in a high environmental risk. To a certain extent, the number and area of water bodies reflect the ability to resist water quality deterioration in catchments. On the other hand, rivers are important channels for pollutants transfer, and SC can reflect the ability and possibility of pollutant transmission in water bodies [55,72]. The close relationship between the SCCs and water quality is reflected in the role of flow paths, water delivery patterns, and the hydrological cycle upstream and downstream of pollutant transfer [11,16].

The correlation between the WER and SC is very complex. In this study, $S_r$ showed a negative relationship with WER both at the national and regional scales of climate

and human activities, indicating that the increase in hydrodynamic forces was helpful in reducing the WER. This result is supported by those of the earlier studies. A case in South Korea suggested that $S_r$ was always negatively associated with total phosphorus at both sub-watershed and buffer scales [18]. You et al., stated that better hydrodynamic conditions were conducive to the migration of pollutants, i.e., catchments with steeper slopes usually had better water quality [30]. The Pearson correlation coefficients of $R_d$, $R_w$, $C_l$, $C_c$, and $R_{nc}$ with WER were both positive and negative in different zones. The $C_l$ reflected the impact of dams on water quality. A case in Turkey showed that some nitrate and phosphate values decreased significantly after dam operation, while other pollutants did not change significantly [73]. Dams also had adverse effects on water quality, for example, Dębska et al., found that water in the Utrata River below the dam reservoir showed higher values of chemical oxygen demand ($COD_{Mn}$) than that above the reservoir [74]. Simultaneously, it was generally believed that higher $R_d$, $C_c$, and $R_{nc}$ created a greater possibility for the transfer of water pollutants and was beneficial to water quality. However, our results showed that $R_d$, $C_c$, and $R_{nc}$ did not always have a beneficial impact on water quality, with a positive Pearson correlation coefficient in some zones, which indicated that the water quality became worse with increasing $R_d$, $C_c$, and $R_{nc}$. The response relationship is complex. Haidary et al. reported that $NO_2^-$ concentration was significantly positively correlated with river density [75], and Deng stated that the concentration of several pollutants showed a negative relationship with river density, while that for others was positive [11]. For $C_c$ and $R_{nc}$, earlier studies reported that the relationship between connectivity and water pollution level was both positive and negative [14,56,76]. The water surface ratio was an important factor for water quality at the catchment scale or smaller spatial scale [11,77], although it showed a weak correlation with the WER in this study at the national and regional scales. $R_w$ was the dominant explanatory variable at the 100 m buffer within a 1 km upstream scale in the Han River basin, South Korea [18]. Deng stated that the influences of different water body types on water quality were discrepant, which was caused by the different purification processes of rivers and lakes [14]. The impact of water body type may be overlooked at larger spatial scales, such as national or regional scales, and should be further studied.

*4.3. Regional Variation of the Influences of Climate and Human Activities*

Earlier studies reported that climatic and anthropogenic factors have a significant impact on the water quality [27,28,78]. Table 7 showed the *r* and *GRAI* between WER and climate, and WER and human activities. The results confirmed the important impacts of climate and human activities on WER. Compared with the value of *r* and *GRAI* between WER and each SCC, the results indicated that the influences of climate and human activities were relatively moderate. In this study, the different relationship between WER and SC were shown in each climate and human activity intensity zone. There was a significant difference in the importance of $S_r$ and $R_w$ in climate zones (i.e., $S_r$ was the most important in humid zones, while it was the least important in other zones, and $R_w$ was the most important in more arid regions). $R_d$ and $C_c$ were more important in the humid and semi-humid zones. A possible explanation is that, compared with the carrying capacity of water for pollutants, the migration level of pollutants had a greater impact on WER for regions with abundant water. Contrastingly, for regions with relatively scarce water resources, the increase in water resource quantity had a greater impact on water pollution control [79] and thus, $R_w$ was more important in these regions. Similarly, the construction of dams and other water conservancy projects altered the fluxes of rivers, affected the self-purification capacity of water bodies and the allocation of water resources [55,73,80,81], and played a key role in semi-arid and arid zones. The dams and reservoirs could create serious damage to river connectivity [82] and could reduce the water for ecological restoration, and the decrease in water resources resulting in water quality deterioration [79]. $R_{nc}$ had low importance in all climatic zones, which was similar to that of the findings in the Southern Jiangsu Plain [11].

**Table 7.** Pearson correlation coefficient (*r*) and gray relational degree index (*GRAI*) between WER and climate, and WER and human activities nationwide.

|  | Climate | Human Activities |
| --- | :---: | :---: |
| *r* | −0.29 * | 0.47 * |
| *GRAI* | 0.48 | 0.60 |

Note: * indicate $p < 0.05$.

China has experienced dramatic rapid development, and high urbanization and intense human activities have influenced the natural structure and connectivity of stream networks and water quality [83–85], thereby affecting the relationship between WER and SC. In the intense and moderate human activity intensity zones, the importance rank of SCCs was significantly different from those of the weak zones, and the *GRAI* value was smaller. It indicates that human activities may have weakened the effects of SC on WER, which is the same as that in earlier studies [86]. More importantly, our results further clarified that the correlation between WER and SC was not weakened monotonically with an increase in human activity intensity, but first weakened and then enhanced. Simultaneously, Liu et al. and Gao et al. found that human factors have a greater impact on water quality than those of natural factors in the Taihu Basin with a high intensity of human activities [86,87]. Therefore, it is reasonable that $R_{nc}$, $R_d$, and $C_c$, which are highly related to human activity intensity (Figure 7), have relatively important effects on WER in intense and moderate zones. The spatial distribution of zones with weak human activity intensity was similar to that of arid zones, showing a similar relationship.

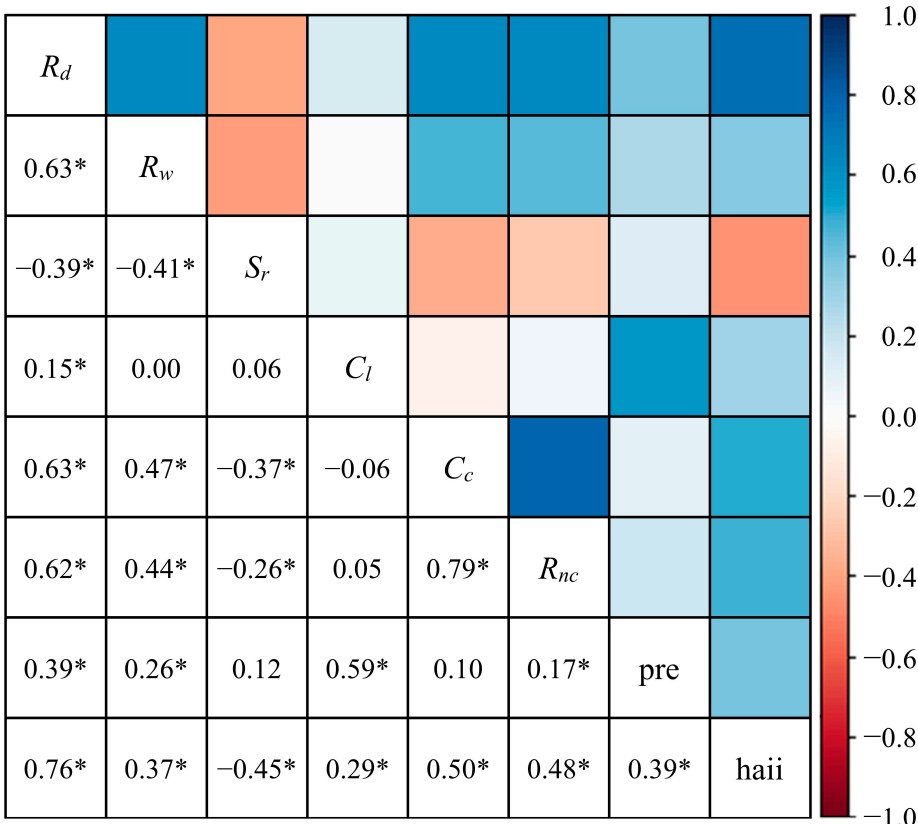

**Figure 7.** Plots illustrating the Pearson correlation coefficients among SCCs (i.e., $R_d$, $R_w$, $S_r$, $C_l$, $C_c$, and $R_{nc}$), precipitation (pre) and human activity intensity index (haii). And * indicate $p < 0.05$.

In this study, the WER was estimated using WPL, and its response to SC was analyzed using correlation analysis and GRA at the national and regional scales. Most studies used

the concentrations of several pollutants to analyze the effect of SC [11,14,16], while an *RI* was used in this study, which was calculated using the water pollution levels of nitrogen and phosphorus. However, these pollutant concentration data were difficult to obtain nationally; thus, the *RI* provided a simple and accessible method to describe water quality and its response to SC. Earlier studies confirmed that the correlation between WER and SC was affected by the study scale, and that the main driving factors varied at different spatial scales [25,72,88]. Compared with earlier studies focusing on catchment scale or even smaller scale, this study analyzed the relationship between WER and SC on a larger scale (i.e., national scale, climate zone scale, and human activity intensity zone scale). Additionally, the relationship between WER and SC may be nonlinear; therefore, our results analyzed by the GRA may be more reasonable than those using the linear analysis method [11,25,89]. However, there was a strong correlation between the SCCs, climate, and human activity intensity (Figure 7), which may lead to inaccurate importance order identification. Many studies suggested that machine learning was an effective method to address the collinearity and nonlinearity issues and was applied to quantify the contribution of driven factors in many study fields [50,90,91]. However, as a data-driven method, machine learning lacks consideration of physical mechanisms. Thus, combination of machine learning with other physically-based methods could be and an effective and complementary way to better quantitatively identify the importance of influencing factors on WER in future research.

## 5. Conclusions

This study indicated that the WER was closely related to the SC, and confirmed the important effect of climate and human activities on the relationship between WER and SC. The relationship between WER and SC showed remarkable regional variability due to the significant impact of climate and human activities. This study systematically quantified the relationship between WER and SC on large spatial scale and emphasized the important role of regional analysis in understanding the complex relationship between WER and SC, especially under different climate and human activity intensity zones. Moreover, this study further clarified that the increase in human activity intensity would not continuously weaken the correlation between WER and SC, but first weakened and then enhanced. However, as the WER is affected by the co-impact of SC, climate, and human activities, a systemic understanding of the relationship between WER and SC requires further consideration of the nonlinearity and collinearity among driving factors. The results emphasize the important role of SC in the WER and the regional variability of their correlation in China. This study highlights that local conditions should be fully considered in WER management and differential management policies should be proposed for various catchments.

**Author Contributions:** Conceptualization, M.C. and L.C.; Methodology, M.C., Z.X., L.C., Q.H. and S.Q.; Formal analysis, M.C., L.C. and Q.H.; Visualization, M.C.; Data curation, Z.X. and S.Q.; Writing—original draft, M.C.; Writing—review & editing, M.C., Z.X., L.C. and Q.H.; Supervision, L.C., Q.H., P.L. and S.Q.; Funding acquisition, L.C. and P.L. All authors have read and agreed to the published version of the manuscript.

**Funding:** This study was supported by the National Natural Science Foundation of China (41890822; 51961145104).

**Data Availability Statement:** Data used in this study are all publicly available and related references are provided in the manuscript.

**Acknowledgments:** The authors appreciate reviewers and editors for their constructive comments that greatly helped to improve the study. We would also like to thank the datasets used in this study.

**Conflicts of Interest:** The authors declare no conflict of interest.

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
