# Peer review of "Exploring the Influences of Stream Network Structure and Connectivity on Water Environment Risk in China"

_water, doi:10.3390/w14244007_

Round 1

Reviewer 1 Report

The manuscript is well written and contains an extensive analysis of statistical dependencies between hydrological spatial characteristics (such as stream network continuity) and environmental risk in terms of water quality. The strength of the work and the main novelty is that the analysis covers entire China, extending to very different regions within the country.

The weakness in my opinion is the observation that the authors seem to overemphasize the role of stream network structure on WER. After reading the results, I get the impression that climate and human activities (including land use by humans) are the two main factors influencing WER. This is obvious, as without precipitation there hardly any high stream flows, and without human activities WER becomes minimal. In the same way, climatic extremes and intensive human disturbance primarily lead to elevated risks in surface water quality. We know that nutrient loads are very low from catchments with minimal human impacts. While the dominant role of climate and human activities becomes clear in results of this study as well, the role stream network structure and connectivity seem to have a secondary role. In fact, the results about the role of different stream characteristics seems to be partly inconclusive.

Another comment is about the methods. I think that the methods seek for relationships and similarity between variables, but the methods do not reveal causal relationship. This could be noted when documenting the results. In the discussion part, one can then continue to make assumptions about causality.

The manuscript is logical and clear, with the exception of the commented issues mentioned above. The literature referred is abundant, but very much focused on studies about China.

I recommend publication after minor revision.

Specific comments

Lines 84-85:

Define what level III catchment scale means.

Equations 1-3:

Provide units of all variables included in the equations.

Line 170:

The term “slope of the i-th grid” is not clear.

Line 171:

The term “total number of the rivers” is not clear. What rivers? On what basis are the rivers defined and separated from each other?

Lines 234 and 236:

You mention river basins, such as Hai, Liao, and Songhua. Show the location of the river basins (that are mentioned in the text) also on the map.

Lines 251-252:

You state that “connectivity …was …worse”. Is “worse” ok term here? Do you really say that “connectivity is bad/worse” or should you state “connectivity is weak/weaker? See also line 253.

Line 268 and onwards:

You have a section labelled as “influences of SC on WER”. Your analysis is not a causal analysis. Analysis is for detecting correlation between variables. Can you document results without statement about causality/influence, but plainly documenting the results of correlation analysis. In the discussion you may discuss causality and influences, some of which may be obvious.

Lines 288-291:

The sentence “Additionally, …” is not clear. Consider revising.

Lines 308-309:

The dependency from climate and human activities seems to be a very important aspect. This should be also addressed better in the abstract.

Tables 4-5:

The results are basically ok. To me it looks like the correlations show inconsistent/inconclusive results about SCC relations with WER. Maybe the average slope in the analysis results is an exception. Does this mean that climate and human activities are in fact the main factors and stream network characteristics have only secondary importance? Should you present the correlation between human impact / climate and WER?

Table 5:

There is correlation coefficient between WER and river density Rd, which is negative for all human activity intensity zones. But then the correlation is positive nationwide. How is this possible? Is there confusion here?

Line 332 and onwards:

What results (Table or Figure) do you explain in this paragraph? Make a reference to Table/Figure.

Line 345:

What is “etc.”. What other spatial differences can there be?

Line 361-362:

I do not agree with this statement. Yes, some relationships can be found, but not consistently and not for all cases.

Lines 370-371:

Is the relationship rather occasional than “close”?

Lines 374:

I agree with this. The correlation between WER and SC seems to be complex. Focus more on this in the discussion and do not try to over-rate the SC correlation.

Lines 388-389:

This statement about “detrimental impact .. nationwide” seems to be an overstatement. Can you rather discuss about tendency, because this generalization does not seem to fair, it overlooks the main factors (climate, human impacts), and it disregards the local characteristics that may matter.

Lines 422-424:

You seem to have many findings that repeat what is already found in earlier studies. Can you be more clear in pointing out your novel added value on the WER issue.

Lines 455:

I think WER is not “closely” related to SC. You should clearly note the climate and human activities at the same time, as you try to do in the second statement of the conclusions, I think. Use some other term than closely, because the close relationship of SC and WER was rather occasional.

Reviewer 2 Report

The article entitled: Exploring the influences of stream network structure and conectivity on water environmental risk in China"

 It is well set up but some sections need to be corrected

The introduction is weak. The literature review should be strengthened and newer references should be used. State the literature challenge and state how you are going to solve that challenge in the region.

The innovation of the research should be expressed.

In the methods section, the climate classification method is not specified. Discussion and conclusion are also weak. Strengthen conclusion and discussion

Reviewer 3 Report

Based on the results shown by the submitted manuscript, it could be reconsidered for publication if you should be prepared to incorporate the major revisions. The reasons for this decision are explained below with general comments:

The abstract should strengthen. I suggest defining the problem/research questions at the beginning without going deeper into the results. Why is important your proposal?

Moreover, although the introduction is well-described, and the focus of this study is clear enough, the list of references should be enlarged. Thus, I suggest improving the introduction, by adding some specific recent references in a broader context of the international literature available on the same topic. Moreover, deeper information about case study is needed (par. 2.1). Regarding the results, they are sufficiently discussed. Finally, make also clear what is novel in your approach in the conclusion and what it could make it more than of regional interest you mentioned.

Moreover, I don’t believe your recommendation to better quantitatively identify the importance of influencing factors on WER in future research with machine learning methods (451-453). In fact, although the machine learning approaches are commonly described in the literature as “sophisticated techniques”, they roughly consider the problem physics. This aspect could be considered a weak point, as process understanding is not the primary target of machine learning methods, so their use may lead to misleading interpretations.

Round 2

Reviewer 2 Report

Dear authors

In figure 7, please show 3 variables

Figure 1 (b) is very low resolution.

In table 2, please add reference for climate classification method

Reviewer 3 Report

The Authors have responded to all requests from reviewer. After the revision process, the paper is

improved and can now be considered for publication. 

Author Response

Thanks very much for your consideration and thorough review of our manuscript.